# The Effect of Forgetting Strategies on Memory Performance: Behavioral and Electroencephalography Evidence

**DOI:** 10.3390/brainsci15121335

**Published:** 2025-12-15

**Authors:** Chenyu Pan, Fuhong Li

**Affiliations:** School of Psychology, Jiangxi Normal University, Nanchang 330022, China; chenyupan@jxnu.edu.cn

**Keywords:** memory, directed forgetting, forgetting strategy, EEG

## Abstract

**Background/Objectives:** This study aimed to examine the effect of different forgetting strategies on intentional forgetting, specifically comparing the passive decay strategy (‘forgetting by keeping the mind blank’) and the active rehearsal strategy (‘forgetting by rehearsing other words’). **Methods:** An item-method directed forgetting paradigm was used in a between-subjects design while the electroencephalogram (EEG) was recorded. **Results:** Behavioral results showed that both strategies produced a robust directed forgetting (DF) effect, but participants in the active rehearsal group recognized more to-be-remembered (TBR) words. Event-related potential (ERP) results indicated that both groups exhibited a DF effect in the cue-induced P2–P3 complex. Compared to the passive decay group, the active rehearsal group did not show a DF effect in the cue-induced later positive component (LPC); instead, a significant DF effect appeared in the P600 during the test phase. Time–frequency results showed that the passive decay group exhibited a significant DF effect in the 9–25 Hz frequency band during the late stage of cue processing, while the active rehearsal group showed a reversed DF effect in the 8–16 Hz frequency band during the mid-stage of cue processing. **Conclusions:** These findings indicate that forgetting strategies do not affect the recognition performance of to-be-forgotten (TBF) words. The active rehearsal strategy led participants to shift attention from TBF to TBR words, resulting in better TBR recognition performance in this group.

## 1. Introduction

Forgetting is often viewed as an undesirable failure of the memory system, such as when a person cannot recall a new colleague’s name or misses an important meeting. However, forgetting can also be advantageous, as it allows individuals to discard outdated information and let go of traumatic experiences that may be better forgotten. Many researchers argue that forgetting is not simply a memory error; rather, it can serve as an adaptive, strategic process [1,2].

Previous studies demonstrated that people can forget information when instructed to do so (for a recent review, see [3]). The item-method paradigm was developed to study intentional forgetting in the laboratory. In this paradigm, participants are presented with items and then receive cues indicating which items to remember and which to forget. The most prominent finding is poorer memory performance for to-be-forgotten (TBF) items compared with to-be-remembered (TBR) items, known as the directed forgetting (DF) effect [4,5,6,7].

As the DF effect has received increasing attention from researchers in recent years [6,7,8,9,10,11,12,13,14,15,16], an important yet often overlooked issue is how forgetting strategies affect the DF effect. Gamboa et al. [17] examined mindfulness as a forgetting strategy (i.e., focusing on breathing) in the context of the DF effect. The results showed that the mindfulness group exhibited a reduced the DF effect compared to control group (i.e., no specific forgetting strategies were instructed), manifested as enhanced memory for TBF items. Recently, Hubbard and Sahakyan [13] examined the effectiveness of forgetting strategies by explicitly instructing participants to use two approaches: the directed forgetting strategy (i.e., attempt to forget) and the thought substitution strategy (i.e., imagining content unrelated to the experiment). They found that these strategies produced different magnitudes of the DF effect and relied on different neural mechanisms. Specifically, the directed forgetting strategy resulted in a larger DF effect and poorer TBF memory performance compared to the thought substitution strategy. However, it is important to note that the thought substitution strategy in this study may not have represented a true forgetting strategy. Participants were told that the words presented before the imagine cue would still be tested, rather than instructed to forget them. This instruction may have led participants to try to remember these words, which could have affected the magnitude of the DF effect observed with the thought substitution strategy.

Additionally, by utilizing the list-method directed forgetting paradigm, Foster and Sahakyan [18] found that without specific instructions, participants spontaneously used various forgetting strategies. Their post-experiment inquiries identified at least three strategies: stopping rehearsal of the TBF items, engaging in thoughts unrelated to the experiment, and doing nothing. Thus, when investigating the effects of forgetting strategies, it is crucial to explicitly specify a particular strategy.

The present study aimed to investigate how different forgetting strategies affect the DF effect and its neural mechanisms using an item-method paradigm with a between-subjects design. Participants were explicitly instructed to use one of two forgetting strategies when presented with a forget cue. The passive decay group was told to forget TBF words by “keeping your mind blank and refraining from rehearsing previous TBR words,” while the active rehearsal group was instructed to forget TBF words by “rehearsing previous TBR words.” We predicted that both groups would show the DF effect; however, memory performance for TBR items would be higher in the active rehearsal group because of extended rehearsal time, while memory performance for TBF items would be lower in the active rehearsal group because of the additional need to actively inhibit TBF items. To investigate the neural mechanisms underlying these strategies, we recorded EEG throughout the experiment. We hypothesized that, in the passive decay group, forget cues would elicit lower neural activity than remember cues because of blank mind maintenance. In contrast, in the active rehearsal group, forget cues were expected to evoke similar or higher neural activity compared to remember cues, because of the increased need to inhibit the current TBF word before rehearsing prior TBR words.

This investigation contributes to the literature on intentional forgetting by providing novel insights into two key aspects. First, it directly compares the behavioral and neurophysiological outcomes of two distinct, instructed forgetting strategies—passive decay and active rehearsal—within the same experimental framework, a comparison that has been scarcely investigated. Second, by employing ERP and time-frequency analysis, it elucidates the distinct neural mechanisms underlying these strategies. The remainder of this article is structured as follows: Section 2 detail the participants, design, procedure, and data analysis. Section 3 presents the behavioral, ERP, and time-frequency findings. Section 4 interprets these results, considers their implications, acknowledges limitations, and suggests directions for future research. Finally, Section 5 summarizes the main findings of the present study.

## 2. Materials and Methods

Participants: The sample size was determined a priori using G*Power 3.1.9.7. Based on a previous meta-analysis [19] which reported a large DF effect size (d = 1.71) for the item-method paradigm in younger adults, a power analysis (α = 0.05, power = 0.95) indicated that a minimum of 7 participants per group would be sufficient. However, to account for potential variability and align with common practice in EEG studies, we recruited a larger sample of participants per group. Sixty-eight native Chinese-speaking college students were recruited and randomly assigned to two groups. Three participants were excluded because they did not follow instructions (specifically, one misunderstood the response rule in the study phase, one reversed the ‘old’/‘new’ response keys during the test phase, and one responded randomly), and five were excluded due to excessive artifacts (>25% of trials) during EEG recording. Thus, data from 60 participants were included in the analyses (30 per group; 28 men and 32 women; mean age = 19.8 ± 1.57 years). All participants were healthy, right-handed, had normal or corrected-to-normal vision, and were not color blind. All participants received payment upon completing the experiment.

Materials: One hundred eighty Chinese double-character nouns were selected from the top 8000 words in “The Modern Chinese Frequency Dictionary,” with a mean frequency of 3.33 per thousand and a mean of 16.74 strokes. The words were divided into two lists of 90: one list contained only nouns referring to objects larger than a computer monitor (e.g., lion), and the other contained only nouns referring to objects smaller than a computer monitor (e.g., butterfly). For each participant, 60 words were randomly selected from each list, resulting in 120 study words. During the test phase, all 180 words were presented (120 old words and 60 new words).

Procedure: The experimental procedure is shown in Figure 1. In the study phase, each trial began with an 800 ms blank screen, followed by a single word displayed inside a black border at the center. Participants judged whether the word referred to an object larger or smaller than a computer monitor. This incidental encoding task ensured attention to the stimuli. After the judgment or after 4000 ms, the word disappeared, and the black border remained for 1000 ms. The border then changed to red or blue with equal probability and stayed on screen for 3000 ms. Participants were told that a red border signaled they should remember the word for a later memory test, while a blue border indicated they should forget the word, as it would not be tested. Color assignments were counterbalanced across participants. Each participant received explicit instructions regarding their assigned forgetting strategy before the experiment. The passive decay group was instructed: “Forget the TBF words by keeping your mind blank and DO NOT rehearse any previous TBR words.” The active rehearsal group was instructed: “Forget the TBF words by rehearsing the preceding TBR words.” During the test phase, participants completed a recognition task in which they indicated whether each presented word was old or new. They were instructed to respond “old” if they remembered the word as previously shown. Each word appeared centrally for 5000 ms, and participants were asked to respond as quickly and accurately as possible. Responses (“old” or “new”) were made using keyboard presses with either the right or left hand, and the assignment of response to hand was counterbalanced across participants.

Behavioral analysis: All behavioral analyses were conducted using R [20]. Recognition accuracy was analyzed using generalized linear mixed models (GLMM), and reaction time (RT) data were analyzed with linear mixed models (LMM). These models predicted participants’ responses based on trial-level behavioral data. Models were fit by maximum likelihood with the lme4 package [21]. To examine differences in recognition by group and word type, group, word type, and their interaction were included as fixed effects. Random factors included intercepts for items, as well as slopes and intercepts for participants for the fixed effect of cue condition. For all statistical tests, the threshold for statistical significance was set at *p* < 0.05.

EEG recording and pre-processing: Brain electrophysiological activity was recorded using a 64-channel EEG system (Brain Products, GmbH, Bavaria, Germany) following the international 10/20 positioning system. The ground electrode was placed at AFz, and the online reference electrode at FCz. Vertical electrooculography (EOG) was recorded with electrodes positioned below the right eye. All interelectrode impedance was kept below 10 kΩ. EEG and EOG signals were amplified with a 0.05–100 Hz bandpass filter and continuously sampled at 1000 Hz. Offline pre-processing and analysis of the EEG data were conducted in MATLAB 2021a using EEGLAB [22] and ERPLAB [23]. After offline re-referencing to the average of the left and right mastoid electrodes, each raw EEG time series was filtered with a 0.1–40 Hz Butterworth filter with a 36 dB/octave roll-off. Filter parameters were selected a priori to remove low-frequency drifts without introducing artifacts in ERP analyses [24], and to eliminate high-frequency noise while preserving beta band activity for time-frequency analyses. In line with Luck’s recommendation [25], we conducted an ICA procedure before segmenting the data into epochs. Components related to eye activity were removed. The time series was segmented into long epochs for time–frequency transformation, spanning −1000 to 3000 ms relative to the onset of each cue in the study phase and each word in the test phase. The EOG-cleaned data were then manually inspected to remove any epochs with remaining artifacts. Overall, data quality was high, with few trials removed (mean 3.2% across subjects).

ERP analysis: Before averaging, trials were baseline corrected using the 200 ms pre-stimulus interval. Because our experimental design uniquely specified explicit forgetting strategies, we did not select predefined channels or time windows for ERP analysis. Instead, we used cluster-based permutation tests [26] to identify significant differences between conditions across the entire data set. This method increases statistical power, controls the Type I error rate [27], and reduces potential biases from subjective channel or time-window selection, which can produce spurious results [28]. Consequently, ERPs within the 0–3000 ms window after cue presentation during the study phase and within the 0–1000 ms window after word presentation during the recognition phase were submitted to permutation testing. In these tests, *t*-tests were calculated at each time point and electrode. Spatiotemporally adjacent significant *t* values were grouped into clusters, with each cluster’s test statistic defined as the sum of its *t* values (hereafter denoted *c*). To determine significance, observed cluster statistics were compared with a permutation null distribution generated by randomly shuffling condition labels 2000 times, recalculating clusters, and extracting the maximum cluster sum statistic for each permutation. The *p*-value for an observed cluster indicates the proportion of permutations in which the maximum cluster statistic exceeded the observed cluster sum. The cluster-based permutation tests were conducted using Brainstorm toolbox [29].

Time-frequency analysis: The Brainstorm toolbox [29] was used to conduct time–frequency analyses. EEG data for each epoch were processed with Morlet’s wavelets. Time–frequency transformation was first performed on individual trials, then averaged across trials for each condition. Wavelets were constructed using Brainstorm’s method, defining each as a scaled and shifted version of a mother wavelet with a temporal resolution of 3 s at 1 Hz. Frequencies for computation ranged from 2 Hz to 30 Hz. Time–frequency power within 0–3000 ms was normalized using decibel (dB) conversion, with −200 to −50 ms as the baseline. Epochs were cut to −200 to 2500 ms to avoid edge effects. All statistical analyses of time–frequency data were conducted using non-parametric cluster-based permutation tests [26]. Data were down sampled to 500 Hz before permutation tests to reduce computational demands.

## 3. Results

### 3.1. Behavior Results

Table 1 presents recognition performance (for raw data and R script, see Appendix A). A GLMM predicting memory accuracy showed significant main effects of Word type (TBR/TBF) in both groups: memory accuracy was significantly higher for TBR words than for TBF words in both the passive decay group (*β* = 0.495, *z* = 4.091, *p* < 0.001) and the active rehearsal group (*β* = 1.082, *z* = 8.540, *p* < 0.001). The Group × Word type interaction was also significant (*β* = 0.586, *z* = 3.384, *p* < 0.001). Follow-up analyses showed that accuracy for TBR words was significantly higher in the active rehearsal group than in the passive decay group (*β* = 0.531, *z* = 2.841, *p* = 0.023), while accuracy for TBF words did not differ significantly between groups (*β* = −0.056, *z* = −0.399, *p* = 0.978).

For reaction times to correctly recognized words, a LLM showed significant main effects of Word type and Group. Participants responded significantly faster to TBR words than to TBF words (*β* = −33.74, *t* = −2.06, *p* = 0.04). Responses were also significantly faster in the active rehearsal group compared to the passive decay group (*β* = −86.47, *t* = −2.07, *p* = 0.04). The Group × Word type interaction was not statistically significant.

### 3.2. EEG Results

For the study phase, the ERPs elicited by cues are presented in Figure 2. In the passive decay group, cluster-based permutation tests comparing R and F cues identified two significant clusters. In the first cluster (Figure 2a), F cues elicited significantly lower amplitudes than R cues between 270 and 440 ms over central-posterior channels (*c* = 13,679, *p* = 0.05). In the second cluster (Figure 2b), F cues elicited significantly lower amplitudes than R cues between 470 and 2200 ms over right frontocentral channels (*c* = 138,919, *p* < 0.01). In the active rehearsal group, one significant cluster was found (Figure 2c), with F cues eliciting significantly lower amplitudes than R cues between 220 and 410 ms over central-posterior channels (*c* = 17,341, *p* = 0.02). In summary, both groups showed reduced amplitudes for F cues over central-posterior regions during 220–440 ms, while the passive decay group exhibited a sustained DF effect from 470 to 2200 ms.

During the test phase, we compared correctly recognized old words (TBR-R or TBF-R) with correctly rejected new words (NEW-C). The cluster-based permutation tests revealed significantly higher amplitudes for both TBR-R and TBF-R words compared to NEW-C words roughly in the 400–600 ms window over central–posterior channels (Figure 3). These effects were significant in both the passive decay group (TBR-R vs. NEW-C: *c* = 28,523, *p* < 0.01; TBF-R vs. NEW-C: *c* = 21,718; *p* < 0.01; Figure 3a) and the active rehearsal group (TBR-R vs. NEW-C: *c* = 29,521, *p* < 0.01; TBF-R vs. NEW-C: *c* = 13,831, *p* < 0.01; Figure 3b). When directly comparing TBR-R and TBF-R words, a significant cluster appeared only in the active rehearsal group, spanning 510–660 ms in parieto–occipital channels, with TBR-R words eliciting more positive amplitudes than TBF-R words (*c* = 6862, *p* = 0.04; Figure 4).

Figure 5 presents the oscillatory activity elicited by cues in study phase. In the passive decay group (Figure 5a), cluster-based permutation tests showed that F cues evoked significantly lower power than R cues over right posterior channels between 870 and 2500 ms (*c* = 637,635, *p* < 0.01), within approximately 9–25 Hz. In the active rehearsal group (Figure 5b), F cues evoked significantly higher power than R cues over left hemisphere channels between 500 and 1000 ms (*c* = −406,348, *p* < 0.01), within approximately 8–16 Hz. No significant clusters were identified during the test phase.

## 4. Discussion

The present study aimed to clarify how explicitly instructed forgetting strategies—passive decay versus active rehearsal—modulate the DF effect and its underlying neural mechanisms. We hypothesized that both strategies would produce a behavioral DF effect, but with superior memory for TBR items and poor memory for TBF items in the active rehearsal group. Neurophysiologically, we hypothesized that forget cues would elicit lower activity in the passive decay group but similar or higher activity in the active rehearsal group. Our behavioral results partly confirmed these hypotheses: a robust DF effect was observed in both groups, with the active rehearsal group recognizing significantly more TBR words but similar memory performance for TBF words. The EEG findings aligned with our predictions, revealing distinct neural signatures for each strategy during cue processing and retrieval. Thus, our findings demonstrate that explicitly instructed forgetting strategies reliably produce DF effect yet engage fundamentally different cognitive and neural processes.

### 4.1. The Influence of Forgetting Strategies on Memory Recognition

Behavioral results showed that both groups demonstrated a strong DF effect; participants recognized more TBR words than TBF words. Although the active rehearsal group recognized more TBR words than the passive decay group, both groups exhibited similar forgetting rates for TBF words. These findings indicate that instructing participants to forget by rehearsing previous TBR words during the F cue presentation enhanced TBR word recognition but did not affect forgetting. This result aligns with a recent study, which found that explicitly redirecting attention and processing resources away from TBF items may not be sufficient to promote item-method directed forgetting [30].

Recognition RTs were shorter for TBR words than for TBF words, and shorter in the active rehearsal group than in the passive decay group. These results indicate that participants recognized TBR items with greater confidence than TBF items in both groups, and that the active rehearsal group developed stronger memory traces for both word types. This interpretation aligns with previous research, which found that faster recognition RTs are associated with stronger memory traces, higher familiarity, and high-confidence judgments [31,32,33]. One explanation for the stronger memory traces of TBF words is that participants may not have fully disengaged from processing TBF words while rehearsing prior TBR words. This lingering processing was more likely in the active rehearsal group because participants were permitted to rehearse prior TBR words during F cue presentation. This explanation aligns with previous studies showing that longer post-cue durations result in better memory performance for both TBR and TBF words [11,34,35].

### 4.2. EEG Results for the Study Phase

During the study phase, both groups showed similar early-stage ERP components. A central to posterior ERP component differentiated F cues from R cues within the 220–440 ms time window. This ERP effect may indicate modulation of the P2–P3 complex [13]. P2 is associated with early attentional processes [36,37]. The smaller P2 amplitude elicited by F cues compared to R cues may suggest that, when presented with an F cue, participants reduced their attention to the recently presented word and removed it from working memory, which led to subsequent forgetting of the word. This interpretation is supported by a study on directed forgetting in working memory, which reported smaller P2 amplitudes following F cues than R cues, indicating reduced attentional allocation [36]. In addition, P3 amplitude modulations are linked to goal-directed stimulus detection and memory encoding processes [38,39]. In the directed forgetting literature, researchers have suggested that increased P3 amplitudes reflect more intensive rehearsal processes triggered by R cues [40,41,42]. Thus, it can be inferred that, upon presentation of the memory cue, both groups showed early attentional engagement with the previously presented word (indexed by P2 amplitudes) and item rehearsal processes triggered by memory cues (indexed by P3 amplitudes). The equivalent modulation of the DF effect on the P2–P3 complex in both groups indicates that the initial neurocognitive mechanisms were similar for the two forgetting strategies.

After the P2–P3 complex, the passive decay group exhibited a sustained DF effect over right frontocentral channels within the 470–2200 ms time window. The late negative component (LNC) observed in the passive decay group likely reflects cognitive differences between “rehearsing previous TBR words (R cues)” and “keeping the mind blank (F cues).” Specifically, R cues evoked activity consistent with continuous selective rehearsal, while F cues elicited responses associated with a mental blank state (i.e., minimal cognitive engagement). The LNC was not observed in the active rehearsal group, possibly because participants in this group were instructed to rehearse previous TBR words after both R cues and F cues, resulting in similar neural activity for both cue types during the later stage of cue presentation. Therefore, the LNC may serve as an ERP signature indicating the continuous implementation of “keeping the mind blank.”

Consistent with the LNC, time–frequency analyses showed that F cues elicited a sustained reduction in oscillatory activity (9–25 Hz) over right posterior regions compared to R cues in the passive decay group. Previous studies indicated that beta oscillations are associated with maintaining a cognitive set or “status quo” [43,44,45]. Increases in the alpha band are thought to protect memory storage by functionally inhibiting task-irrelevant information or brain regions [46,47], as well as by inhibiting interference from previously encoded information [48,49]. Sustained attention also relies on alpha-mediated suppression of task-irrelevant cortical areas [50]. Therefore, R cues likely elicited higher power in the alpha and beta bands, possibly reflecting that R cues prompted sustained cumulative rehearsal (i.e., rehearsing previous TBR words during cue presentation), which required simultaneous inhibition of unwanted TBF word memories. In contrast, the reduction in alpha and beta oscillatory activity after F cues suggested that “keeping mind blank” involved less inhibition [51], serving as a distinct forgetting mechanism compared to active inhibition.

In the active rehearsal group, the time–frequency results revealed a different cluster compared to the passive decay group. Specifically, F cues elicited higher alpha power than R cues during the 500–1000 ms interval. As previously noted, alpha oscillations indicate active inhibition of local information processing [51,52,53] and are linked to the functional inhibition of task-irrelevant brain regions or distracting neural activity [46,54,55,56]. Fellner, Waldhauser and Axmacher [52] argued that increased alpha power during selective rehearsal may reflect a focus on internal representations, which requires suppressing potentially distracting bottom-up information flow [51,52,57,58]. Therefore, the greater alpha power observed for F cues compared to R cues suggests that participants needed to inhibit the words they had just encountered when shifting attention from TBF words to TBR words. This finding further supports the idea that participants must inhibit F words before rehearsing the preceding R words [59].

### 4.3. EEG Results for the Test Phase

By comparing old words (TBR or TBF) with new words, we found a significant ERP old/new effect in both the passive decay and active rehearsal groups. In both groups, TBR and TBF words elicited higher amplitudes than new words in the later positive component (LPC) time window over the parietal scalp. The LPC old/new effect is associated with the recollection process [60,61,62,63], indicating that participants distinguished both TBR and TBF words from new words through recollection. Notably, previous DF studies [32,63] showed that TBR items evoked LPC old/new effects, whereas TBF items did not. In contrast, the present study found that both TBR and TBF words elicited reliable LPC old/new effects. One possible explanation for this discrepancy is the difference in experimental design: participants completed a word-size judgment task (i.e., judging whether the word was larger or smaller than the computer monitor) before receiving the memory cue. Previous DF studies did not include this intentional encoding task. This task may have promoted more elaborate encoding for both TBR and TBF words, which could have enhanced recollection-based recognition for both word types during the test phase and resulted in the observed LPC old/new effects.

In addition, only the active rehearsal group showed a P600 component in the ERP results that differentiated TBF words from TBR words, indicating that participants in active rehearsal group could distinguish TBF words from TBR words in the brain during the test phase. P600 effects are typically activated during memory tasks and have been associated with accurate source memory [64]. Source memory refers to remembering the contextual information associated with the origin of a specific episodic memory [65]. In the present study, it involved recalling whether an old word was originally assigned to the TBR or TBF condition. The observation that the active rehearsal group had better source memory than the passive decay group aligns with the RTs results, confirming that participants developed deeper memory traces and greater familiarity through the use of an active rehearsal strategy.

### 4.4. Limitations and Future Directions

Although recognition reaction times were slower in the passive decay group, recognition accuracy for TBF words did not differ significantly between groups. One possible explanation is that the incidental encoding task (i.e., judging whether a word was bigger or smaller than a computer monitor) before the R/F cues (for similar setting, see [66]) required participants to process all words at a deeper level, making all words equally difficult to forget, regardless of the forgetting strategy used. Future studies could employ a shallower initial encoding task or vary the encoding depth to better isolate the specific effect of the instructed forgetting strategies.

Furthermore, some participants in the passive decay group reported difficulty maintaining a completely blank mind throughout the cue duration and occasionally thought about the TBR words. Thus, maintaining a blank mind for the entire 3 s cue period appears challenging. Future research could consider shortening the cue duration (e.g., 2 s) to reduce the effect of this factor.

## 5. Conclusions

In conclusion, this study shows that both passive decay and active rehearsal strategies effectively produce the DF effect; however, the active rehearsal strategy resulted in better memory performance for TBR words. Additionally, each strategy is supported by distinct neural mechanisms. The passive decay strategy functions through late-stage cue presentation, as indicated by the LNC and alpha/beta band effects. In contrast, the active rehearsal strategy involves proactively shifting attention from TBF items to TBR items during the mid-stage of cue presentation, as reflected by increased alpha oscillatory activity.

## Figures and Tables

**Figure 1 brainsci-15-01335-f001:**
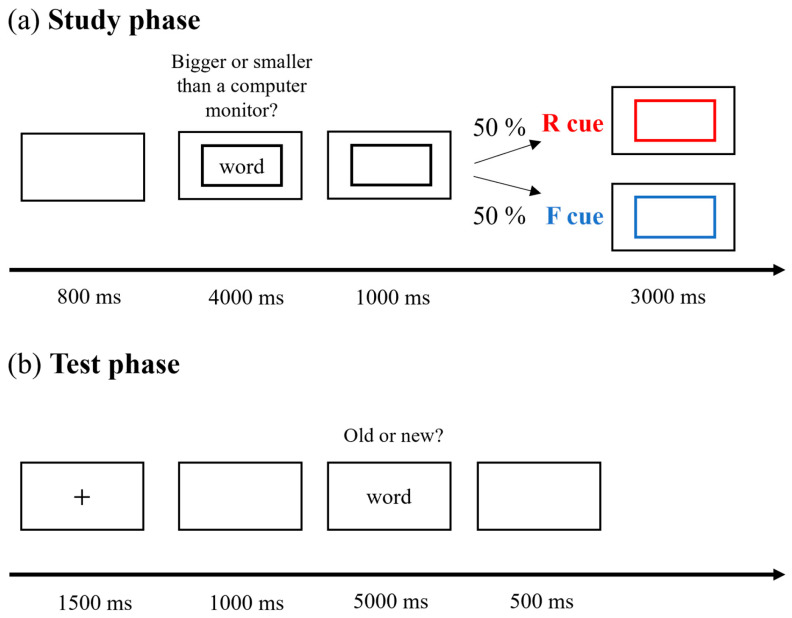
Schematic representation of the experiment. "+" is the fixation point. The red and blue boxes represent the Remember and Forget cues, respectively. R: Remember. F: Forget.

**Figure 2 brainsci-15-01335-f002:**
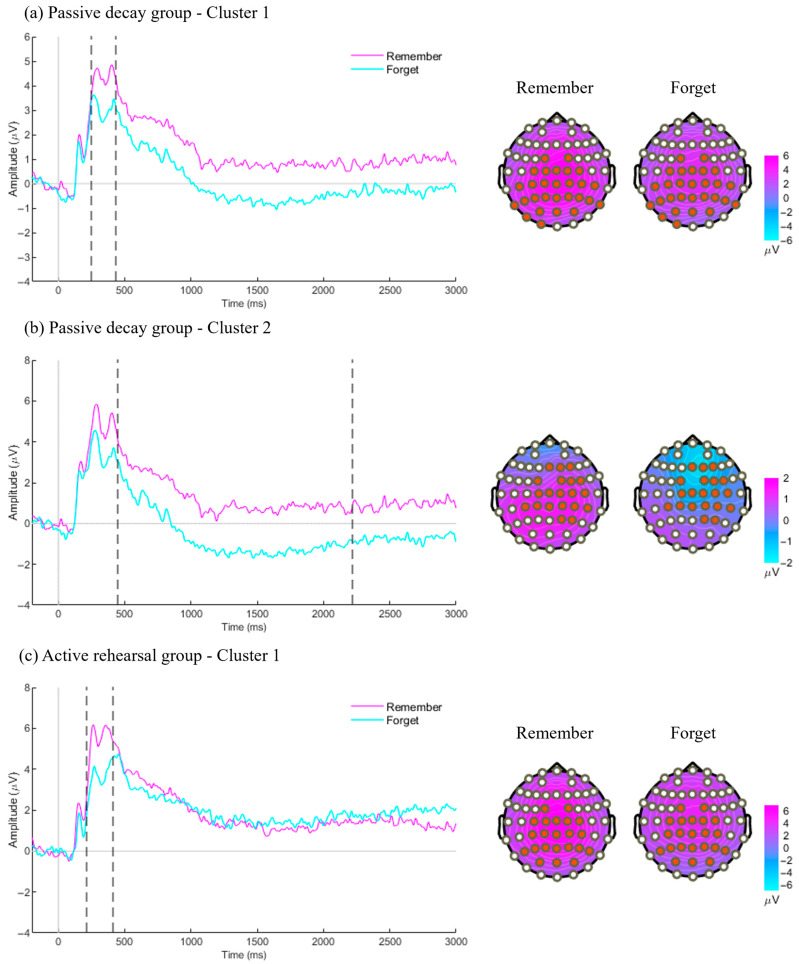
ERP results from the study phase. ERPs were time-locked to the memory cues (“Remember” cue and “Forget” cue). Channel clusters (marked by red dots on topographies) were averaged to generate the ERP plots. Corresponding average topographies, with their time windows indicated by dotted lines on the ERP, are displayed alongside. (**a**) The first cluster identified in the passive decay group. It was located over central-posterior regions and occurred around 270–440 ms. (**b**) The second cluster identified in the passive decay group. It was located over right frontocentral channels and occurred around 470–2200 ms. (**c**) The cluster identified in the active rehearsal group. It was located over central-posterior regions and occurred around 220–410 ms.

**Figure 3 brainsci-15-01335-f003:**
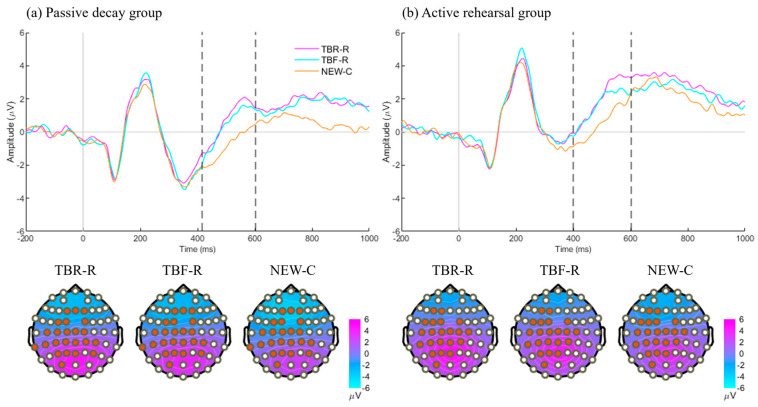
ERP results for the test phase. ERPs were time-locked to the words. Channel clusters (marked by red dots on topographies) were averaged to generate the ERP plots. Corresponding average topographies, with their time windows indicated by dotted lines on the ERP, are displayed in the bottom. (**a**) Passive decay group. Two similar clusters were plotted together, namely, TBR-R vs. NEW-C and TBF-R vs. NEW-C. Both clusters located over central–posterior channels and occurred around 420–600 ms. (**b**) Active rehearsal group. Two similar clusters were plotted together, namely, TBR-R vs. NEW-C and TBF-R vs. NEW-C. Both clusters located over central–posterior channels and occurred around 400–600 ms.

**Figure 4 brainsci-15-01335-f004:**
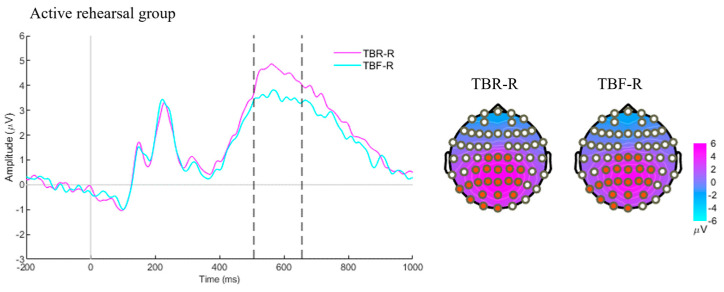
ERP results by comparing TBR-R and TBF-R in the active rehearsal group. ERPs were time-locked to the words. Channel clusters (marked by red dots on topographies) were averaged to generate the ERP plots. Corresponding average topographies, with their time windows indicated by dotted lines on the ERP, are displayed alongside. This cluster spanning 510–660 ms in parieto–occipital channels.

**Figure 5 brainsci-15-01335-f005:**
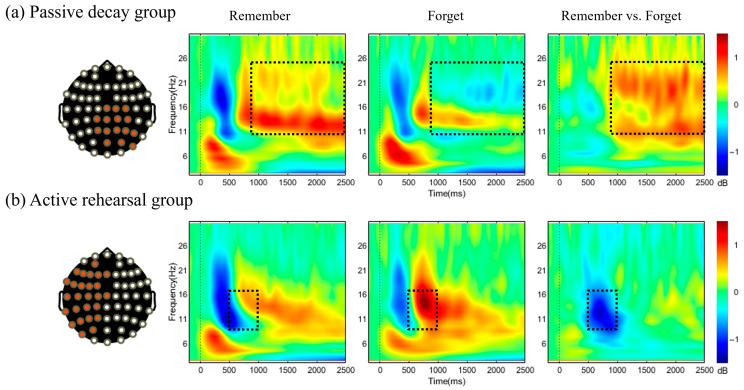
Time-frequency results for the study phase. The channel clusters appeared as red dots on the scalp plot, and the average of these channels was used to generate the time–frequency plot. The dotted lines indicate the moment of cue presentation, and the dotted boxes denote significant clusters. (**a**) The passive decay group identified a cluster over right posterior regions between 870 and 2500 ms, spanning approximately 9–25 Hz. (**b**) The active rehearsal group identified a cluster over left hemisphere regions between 500 and 1000 ms, spanning approximately 8–16 Hz.

**Table 1 brainsci-15-01335-t001:** Mean and standard deviation (SD) of accuracy and RTs data in both groups. Note that RTs data was based on the correctly recognized words. NEW-C refers to correctly rejected new words.

	Accuracy			Reaction Time (ms)		
	TBR	TBF	New	TBR-R	TBF-R	New-C
Passive decay group	0.82 (0.11)	0.76 (0.09)	0.85 (0.08)	931 (190)	962 (177)	1024 (202)
Active rehearsal group	0.89 (0.06)	0.75 (0.09)	0.82 (0.12)	814 (146)	877 (143)	945 (213)

## Data Availability

The raw data supporting the conclusions of this article will be made available by the authors on request.

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
