# Peer review of "The Effect of Forgetting Strategies on Memory Performance: Behavioral and Electroencephalography Evidence"

_brainsci, 2025, doi:10.3390/brainsci15121335_

Round 1
Reviewer 1 Report
Comments and Suggestions for Authors
- Previous studies had demonstrated that people can intentionally forget information; however, few studies examined the effect of forgetting strategies on intentional forgetting. The recommendation is that this sentence be removed entirely from the abstract and replaced with the purpose of the study.
- At the end of the introduction, after the purpose and the hypothesis of the study, it is recommended to include the following:
- The novelty elements presented in this study and its place in the scientific literature;
- The structure of the article by sections.
- It is recommended that authors focus more on the study's inclusion and exclusion criteria.
- (30 per group; 28 88 men and 32 women; mean age = 19.8 years, standard deviation = 57). Here, it is advisable to appear 19.8 ± 1.57.
- The paper does not include data on the statistical testing or the statistical software used.
- It is strongly recommended that the sample size, statistical power, or both be specified in the paper.
- After that, the proposal is to include a normality test so that journal readers can assess the normality of the data distribution.
- Please also specify the statistical significance threshold.
- Previous studies have found that, without specific instructions, participants spontaneously used various forgetting strategies, leading to different levels of the forgetting effect [2,18]. Please start the discussion by stating the purpose and hypothesis of the study, and specify whether the hypothesis was validated and whether the purpose was achieved.
- 4. Limitations. It is recommended that future research directions also be addressed here: 4. Limitations of the study and future research directions.
- Given the journal's prestige, please remove/replace the following outdated bibliographical sources from the study: 4, 20, 30-31, 35-38, 46, 60-64, 67.
Moderate revisions of the English language are required.
Reviewer 2 Report
Comments and Suggestions for Authors
In this study, the authors examined how different forgetting strategies affect memory performance and the underlying neural mechanisms, using behavioral and electroencephalographic (EEG) analysis. Two forgetting strategies were tested: passive decay (keeping the mind blank) and active rehearsal (rehearsing other words). The findings suggest that both strategies produced a directed forgetting effect, but the active rehearsal group recognized more to-be-remembered (TBR) words. These results indicate that active rehearsal enhances TBR recognition by shifting attention away from to-be-forgotten (TBF) words. Overall, the study is well-designed with an interesting topic and hypothesis. However, some points need to be clarified throughout, and methodological details to further improve the manuscript came to mind, as detailed below.
Comment #1: Please explain all abbreviations when they are first mentioned in text. For example, this is not done properly for LPC (later positive component; line 18) and ERP (event-related potential; lines 16, 139, 148).
Comment #2: (Lines 43-67): shorten and merge the paragraphs at (lines 43-67). The authors repeat the same information stated in previous studies (refs #14 and #18) to justify the need for this research. Also, at line 61, the authors claim that “To date, only two studies have examined the impact of forgetting strategies on the DF effect” (i.e., refs #14 and #18). I would recommend avoiding using "only". It can be argued that these are two of the studies directly related to this study, but other studies might be relevant in different contexts for intentional forgetting, as seen through the manuscript and listed in the reference list.
Comment #3: It would be great if the authors could add a brief overview of the main findings of this study and how the results support the hypotheses at the end of the introduction (lines 68-83), beyond the objectives and hypotheses.
Comments #4: (lines 85-89), For the inclusion and exclusion criteria for the students in each group, if I am not mistaken, the supplementary Excel sheet still presents the trials from all participants (i.e., 68) without an excluded mark. Please add an extra column indicating which subjects were excluded and that the groups ended up with 30 per group (line 88).
Comment #5: For behavioral data results, I see the “t” values presented at lines 187-192 and the "z" values in the top paragraph, lines 181-186, for linear mixed-effect models. Furthermore, for ERP analysis, statistical significance was determined using cluster-based permutation testing, with the cluster-level statistic defined as the sum of t-values within each cluster. If possible, specify test values for differences among groups and within clusters in a specific time window (e.g., summed t-value = xx, p = 0.04), which could help clarify the effect sizes of the findings (not just stating p-values). At line 214, I think it should be “p” instead of “ps” or what do you mean by "ps"? Regarding the analysis of the different channel sites and clusters, I feel that conclusions like some effects were found in specific regions or electrodes but not others are not thoroughly clarified. Were channels directly compared?
Comment #6: Figures related: I would suggest altering your color scheme for line plots to avoid red/green pseudocoloring together. Swapping red for magenta, or green for cyan, are easy switches that might improve accessibility to color blind readers, when possible. Additionally, the figure legends contain very limited information about the statistical tests used. For example, if there were significant effects in particular time windows. Specify a letter for each panel, for each group, and for the topographical maps. Then, refer to which panel presents the text of the results, since it is a bit confusing and not specific when just pointing to the whole figures in the manuscript. I would rather also expand some information about the abbreviations, statistical tests used, sample size, time window, plot type, mean, and errors, etc.
Comments #7: References related: Throughout the manuscript, some references do not fully support the described antecedent. For some statements, there are several references; some may be relevant, but others appear to be inappropriately cited given the context. I am not going to point to specific studies to filter out, and it is up to the authors to decide which ones are relevant to cite in their context. Also, I am missing contrast references; I see most of the references provided in the discussion are consistent or align with their findings. Although the authors claimed that “few studies examined the effect of forgetting strategies on intentional forgetting”. Furthermore, the references provided (refs #59 and #65) do not fully support the given argument. For example, citing reference #59 (Lines 325-329) is appropriate if you want to support a claim that an age-related inhibitory deficit exists in directed forgetting. The reference #65 (Lines 346-351) does not specifically support the connection between the P600 component and source memory performance, nor does it address TBF and TBR word distinctions or rehearsal strategies. Maybe it is only useful here for defining source memory.
Round 2
Reviewer 1 Report
Comments and Suggestions for Authors
Congratulations on your hard work in developing this scientific material!
Comments on the Quality of English LanguageModerate revisions of the English language are required.
Reviewer 2 Report
Comments and Suggestions for Authors
I do not have any further comments. In my opinion, all my concerns have been addressed and reflected in the updated version.